# Long-Term Outcomes and Causes of Death among Medullary Thyroid Carcinoma Patients with Distant Metastases

**DOI:** 10.3390/cancers13184670

**Published:** 2021-09-17

**Authors:** Hyunju Park, Heera Yang, Jung Heo, Tae Hyuk Kim, Sun Wook Kim, Jae Hoon Chung

**Affiliations:** Division of Endocrinology & Metabolism, Department of Medicine, Thyroid Center, Samsung Medical Center, Sungkyunkwan University School of Medicine, Seoul 06351, Korea; hj1006.park@samsung.com (H.P.); heera.yang@samsung.com (H.Y.); jung91.heo@samsung.com (J.H.); taehyukmd.kim@samsung.com (T.H.K.); sunwooksmc.kim@samsung.com (S.W.K.)

**Keywords:** medullary thyroid carcinoma, distant metastasis, oncologic outcome, cause of death

## Abstract

**Simple Summary:**

The prognostic significance of metastatic sites, and immediate cause of death in MTC patients with distant metastasis remains unclear. In this study, we calculated the time from metastasis to death, and we found that long-term oncologic outcomes were differ from initial metastatic site; the hazard ratio for bone metastatic sites (HR: 5.42; *p* = 0.044) and multisite metastasis (HR: 6.11; *p* = 0.006) were significantly higher than for lung metastasis. Complications due to the progression of distant metastasis were the most common cause of death, followed by complications related to chemotherapy and airway obstruction.

**Abstract:**

Distant metastasis is a poor prognostic factor in medullary thyroid carcinoma (MTC), but the significance of differentiating the characteristics according to the site of distant metastasis remains unclear. This study aimed to evaluate the clinical characteristics and long-term oncologic outcomes in MTC patients with distant metastasis. We identified 46 MTC patients with distant metastasis between 1994 and 2019. Clinical characteristics were compared based on the timing of the detection of distant metastasis. Additionally, survival rates following the detection of distant metastasis were evaluated to compare the clinical significance of metastatic site. The detailed causes of death were also investigated. Of the 46 patients, 15 patients (32.6%) had synchronous distant metastasis and 31 patients (67.4%) had metachronous distant metastasis. There was no clinical difference between these two groups except regarding initial surgical extent. The lung (52.2%) was the most common metastatic site, followed by the bone (28.3%), mediastinum (19.6%), liver (17.4%), adrenal gland (4.3%), brain (4.3%), kidney (2.2%), and pancreas (2.2%). Patients with bone metastasis and multisite metastasis had significantly worse prognoses than those with lung metastasis (hazard ratio: 5.42; *p* = 0.044 and hazard ratio: 6.11; *p* = 0.006). Complications due to the progression of distant metastasis, airway obstruction due to tracheal invasion, and complications related to chemotherapy were leading causes of death. In conclusion, there was no difference in clinical characteristics according to the timing of distant metastasis. Oncological outcomes differed by metastatic site.

## 1. Introduction

Medullary thyroid carcinoma (MTC) is an uncommon disease derived from the parafollicular cells of the thyroid gland [1]. It occurs either sporadically or in a hereditary form and accounts for 1% to 2% of all thyroid carcinoma cases in the United States and 0.6% of all cases in Korea [2,3]. MTC presents with locoregional metastasis in 50% of patients and distant metastasis in 10% to 15% of patients at the time of initial diagnosis [4]. Surgery represents the sole opportunity for the cure of MTC because MTC cells do not concentrate radioactive iodine and are not sensitive to the manipulation of thyroid-stimulating hormone (TSH) levels. Therefore, the prognosis of MTC is worse than that of differentiated thyroid carcinoma (DTC), and the mortality rate from progressive MTC is reported to be 15.6% [5,6].

The initial standard treatment for MTC consists of total thyroidectomy with central lymph node dissection. Unlike patients with DTC, those with MTC are not eligible to receive radioactive iodine therapy; therefore, MTC with distant metastasis is incurable [7]. Previous reports have observed synchronous distant metastasis (SDM) in 5–8% of patients and metachronous distant metastasis (MDM) in 38% of advanced MTC patients [5,8,9,10,11]. The lungs, liver, and bone are common metastatic sites, and distant metastasis is known to be a poor prognostic factor for long-term oncologic outcomes [9,12,13,14].

However, most previous studies did not differentiate the characteristics according to the site of distant metastasis, and included only a relatively small number of patients with distant metastasis because of the rarity of disease. Additionally, some studies have not considered the timing of the detection of distant metastasis, which can cause lead-time bias. Thus, the prognostic significance of metastatic sites in MTC remains unclear. In addition, the causes of death among MTC patients with distant metastasis have seldom been reported. The aim of this study was, therefore, to evaluate the clinical differences and long-term oncologic outcomes according to the detection time of distant metastasis and metastatic sites. We also analyzed causes of death among deceased MTC patients with distant metastasis.

## 2. Materials and Methods

### 2.1. Study Population

We retrospectively reviewed 246 patients who were treated for MTC at Samsung Medical Center between December 1994 and December 2019. Among the 246 MTC patients, 46 patients with distant metastasis (18.7%) were identified. Most of these patients had undergone total thyroidectomy and central neck dissection. Detailed clinicopathological information was collected by a review of electronic medical records. The cause of death was obtained from the national death certificate data, listed in the Korean National Statistical Office. The institutional review board (IRB) approved this study, and the need for informed consent was waived because this was a retrospective study (IRB No. 2020-05-064).

### 2.2. Definition of Distant Metastasis

We defined SDM as distant metastasis detected before or within six months of the initial surgery and MDM as distant metastasis detected more than six months after the initial surgery, respectively. Distant metastasis was confirmed by pathology when available and/or imaging studies, including computed tomography, magnetic resonance image, positron-emission tomography scans and bone scan.

Therapy for distant metastasis was performed with both local and systemic treatments. Local therapies included metastasectomy, external beam radiation, anti-resorptive agents for bone metastasis, and chemo-embolization, while systemic therapies included conventional chemotherapy before the introduction of tyrosine kinase inhibitor (TKI), and then TKIs such as vandetanib, cabozantinib, lenvatinib, and selpercatinib.

### 2.3. Statistical Analysis

Continuous variables were presented as mean (standard deviation, SD) or median (interquartile range, IQR) values, as appropriate. Categorical variables were presented as numbers (percentages). For the comparison of continuous variables, a paired *t*-test or the Wilcoxon signed-rank test were used. A chi-squared test was used for categorical variables. Kaplan–Meier analysis was performed to calculate cancer-specific survival (CSS) rates, and groups were compared using a log-rank test. A Cox proportional hazard model was used to identify factors associated with CSS, and survival rate from the detection of metastasis. Statistical analysis was performed using the Statistical Package for the Social Sciences version 25.0 for Windows software program (IBM Corporation, Armonk, NY, USA).

## 3. Results

### 3.1. Baseline Characteristics of Study Participants with Distant Metastasis

The clinicopathological characteristics of the 46 MTC patients with distant metastasis are presented in Table 1.

The mean (SD) age at initial diagnosis was 45.9 (15.0) years, and 23 (50.0%) patients were women. Among these 46 patients, three (6.5%) were diagnosed with multiple endocrine neoplasia type 2A (MEN 2A). The median (IQR) primary tumor size was 3.0 (2.0–4.4) cm, and 35 patients (76.1%) had N1b. Forty-one patients (89.1%) underwent total thyroidectomy and central neck node dissection. One patient (2.2%) underwent lobectomy as their first operation, followed by complete thyroidectomy with neck node dissection. The remaining four patients (8.7%) had advanced disease at the time of their diagnosis and were unable to undergo surgery due to the accompanying severe disease. According to the eighth American Joint Committee on cancer/tumor–node–metastasis staging system, one patient (2.2%) had stage I disease, three (6.5%) had stage II disease, three (6.5%) had stage III disease, 20 (43.5%) had stage IVA disease, two (4.3%) had stage IVB disease, and 15 (32.6%) had stage IVC disease. Information about initial TNM stage in two patients were not available.

### 3.2. Comparison of Clinical Characteristics between the SDM and MDM Groups

Table 2 shows clinicopathological characteristics according to the detection time of distant metastasis. Distant metastasis was found at the time of initial diagnosis in 15 patients (32.6%) and during follow-up in 31 patients (67.4%), respectively. Age, sex, tumor type, primary tumor size, cervical lymph node metastasis, and the number of deceased patients were not different between the two groups. However, the initial surgical extent was significantly different between the two groups due to the presence of inoperable patients in the SDM group.

### 3.3. Survival According to Metastasis Site

Metastatic sites are listed in Table 3. The lung (52.2%) was the most common metastatic site, followed by the bone (28.3%), mediastinum (19.6%), liver (17.4%), adrenal gland (4.3%), brain (4.3%), kidney (2.2%), and pancreas (2.2%). In the SDM group, lung metastasis (40.0%) and multiorgan metastasis (40.0%) were most commonly detected, while, in the MDM group, lung metastasis (32.3%) was most commonly detected, followed by mediastinum/multiorgan (each 22.6%), bone (12.9%), and liver (9.7%) metastasis. Multiorgan metastasis was found more frequently in the SDM group than in the MDM group (40.0% vs. 22.6%). Of the 16 patients with lung metastasis, most patients had micronodular (<1 cm) metastasis; only two patients had macronodular metastasis (≥1 cm).

Of the 46 MTC patients with distant metastasis, 19 patients (41.3%) died of MTC during the median follow-up period of 53 months (range: 3–255 months) after their initial operation. The CSS rates of all enrolled patients were 95.7% at one year, 82.2% at three years, 77.4% at five years, and 62.7% at 10 years. However, the CSS significantly differed according to metastatic site (log-rank *p* = 0.001) (Figure 1).

The CSS rates of patients with lung metastasis were 100.0% at one year, 87.1% at three and five years, and 74.6% at 10 years. The CSS rates of patients with mediastinum metastasis were 100.0% at one, three and five years, and 85.7% at 10 years. Those of patients with bone metastasis were 100.0% at one and three years, 75.0% at five years, and 50.0% at 10 years, and those of patients with liver metastasis were all 100.0% at one, three, five and 10 years, respectively. Finally, the CSS of patients with multiorgan metastasis was 84.6% at one year, 53.8% at three years, 46.2% at five years, and 30.8% at 10 years. When analyzing the factors affecting the prognosis, age, primary tumor size (>4.0 cm), the presence of initial distant metastasis, and initial distant metastasis site were significant risk factors for CSS. After adjustment for all other variables, the initial distant metastasis site was an independent risk factor for cancer-specific survival (Appendix A).

Considering the lead-time bias, we evaluated the survival from the detection of distant metastasis to death. Survival rates from the detection of distant metastasis were significantly different according to the metastatic site (log-rank *p* = 0.007) (Figure 2). The five-year survival rates from the detection of distant metastasis were 100.0% for liver metastasis, 74.6% for lung metastasis, 62.5% for mediastinum metastasis, 30.8% for multisite metastasis, and 0% for bone metastasis. The hazard ratio (HR) for metastatic sites was significantly higher for bone metastasis (HR: 5.42; *p* = 0.044) and multisite metastasis (HR: 6.11; *p* = 0.006) than for lung metastasis (Table 4). In fact, four of the 19 patients died of airway obstruction, which is thought to be due to loco-regional disease, although they had distant metastasis. Thus, we performed subgroup analysis with 42 patients, except those who died of airway obstruction. The HRs for bone metastasis (HR: 9.35 and 95% CI: 1.46–60.11; *p* = 0.019) and multisite metastasis (HR: 7.65 and 95% CI: 1.61–36.32; *p* = 0.010) were also significantly higher than those for lung metastasis.

### 3.4. Causes of Death among Deceased Patients

Detailed information for the 19 deceased patients is presented in Table 5. Among these patients, complications due to the progression of distant metastasis (36.8%) were the most common cause of death, followed by complications related to chemotherapy (26.3%) and airway obstruction due to tracheal invasion (21.1%). Complications due to the progression of distant metastasis included respiratory failure due to lung metastasis (15.8%), complications due to bone metastasis (15.8%), and mental changes due to brain metastasis (5.3%). The remaining three deceased patients (15.8%) died of MTC, but the direct cause of death could not be identified. Out of the five patients who received chemotherapy, three patients received vandetanib, one received lenvatinib, and one received conventional chemotherapy.

## 4. Discussion

The present study was designed to evaluate the long-term outcomes of MTC patients with distant metastasis and to identify the causes of death of deceased patients. Several studies have already identified prognostic factors for oncological outcome in patients with MTC [1,4,9,11,13,14]. Survival following MTC diagnosis is significantly associated with age and initial stage, including distant metastasis [7,15,16]. Lee et al. reported that distant metastasis was the strongest predictor of overall and progression-free survival [17]. However, previous studies have included only a small number of MTC patients with distant metastasis or have not focused at all on this population, despite their poor prognosis. Herein, we evaluated a relatively large number of MTC patients with distant metastasis and found that oncologic outcomes varied depending on the metastatic site.

In this study, distant metastasis was recorded in 18.7% (46/246) of MTC patients, with approximately one-third (15/46) of cases confirmed at the time of initial diagnosis and two-thirds (31/46) of cases confirmed during the follow-up period, consistent with previous studies. Machens et al. reported that the distant metastasis rate, regardless of detection timing, ranged from 6% to 23% according to the period of thyroidectomy between 1995 and 2015 [18]. In the present study, the lung (52.2%) was the most common metastatic site, followed by the bone (28.3%), mediastinum (19.6%), liver (17.4%), and other organs (e.g., adrenal gland, brain, kidney, and pancreas). In the case of SDM, lung metastasis (40.0%) and multiorgan metastasis (40.0%) occurred most frequently, with similar frequency rates, whereas, in MDM patients, distant metastasis was found relatively evenly in several organs, although lung metastasis (32.3%) was the most common finding. Interestingly, multiorgan metastasis was confirmed more commonly in patients with SDM than in those with MDM (40.0% vs. 22.6%). Most cases of lung metastasis were multiple and micronodular in nature. These results are consistent with previous studies: Sippel et al. reported that distant spread of MTC frequently involved the mediastinum, lung, liver, and bone [4], and lung metastases were usually multiple in nature [19]. Previously, the liver was reported to be the most frequent site of distant metastasis, occurring in 45% of patients with advanced MTC [7,20], but, in this study, liver metastasis was less frequently seen, and patients with liver metastasis showed an indolent clinical course. Interestingly, all four patients with liver metastasis survived until at least five years after the detection of distant metastasis without any death.

MTC is more aggressive than DTC, but it sometimes behaves as an indolent malignancy. Overall, 10-year survival rates have been reported to vary from 70% to 90% based on the results of several studies [13,21,22]. Furthermore, previous studies reported that MTC patients whose tumors were confined to the thyroid had a 10-year survival rate of 96% [15]; however, the 10-year survival rate decreased to 75% with the existence of loco-regional disease, and to 40% in cases with distant metastasis [13,15]. In the present study, we focused on MTC patients with distant metastasis; therefore, survival rates were evaluated only for MTC patients with distant metastasis, not for all MTC patients. More than forty percent (19/46; 41.3%) of MTC patients with distant metastasis died of MTC during a median follow-up period of 53 months after their initial operation. The CSS rates gradually declined over time, from 95.7% at one year to 82.2% at three years, 77.4% at five years, and 62.7% at 10 years. Additionally, despite the lung being the most common site of metastasis, the CSS rate for lung metastasis was significantly higher than those for bone metastasis and multisite metastasis (10-year CSS: 74.6% vs. 50.0% and 30.8%, respectively).

Unfortunately, there is no curative treatment modality for MTC with distant metastasis. Therefore, the therapeutic goals for these patients are to provide loco-regional disease control and palliate metastatic symptoms that sometimes threaten life. Recently, target therapy with tyrosine kinase inhibitors has been emerging, but the limited efficacy and potential toxicities of systemic therapy should be considered [23,24,25,26]. Thus, clinicians must decide which patients require further therapy. Previous studies have reported CSS rates in patients with SDM of from 50% to 55% at five years and 38% at 10 years [1,8,14]. However, these results were recorded for all MTC patients with distant metastasis, without considering the specific metastatic sites. In the present study, the five- and 10-year CSS rates in patients with lung metastasis were similar to those in previous investigations of all MTC patients (87.1% vs. 85–89% at five years and 74.6% vs. 71–87% at 10 years) [27,28,29,30]. Meanwhile, five- and 10-year CSS rates were 75.0% and 50% in patients with bone metastasis and 46.2% and 30.8% for patients with multisite metastasis, respectively. This finding is clinically important in determining which patients require more aggressive therapeutic approaches to manage distant metastasis.

Considering the lead-time bias, survival time after the discovery of distant metastasis is important for the clinical implications, but previous studies did not consider the detection time of distant metastasis [5,10,13]. Some studies reported survival rates of 25% at five years and 10% at 10 years after the detection of distant metastasis [31], but no recent studies have focused on survival time after the detection of distant metastasis. Therefore, we re-evaluated survival from the detection of distant metastasis to death to overcome the lead-time bias. In the present study, the five-year survival rates from the detection of distant metastasis were 100.0% for liver metastasis, 74.6% for lung metastasis, 62.5% for mediastinum metastasis, 30.8% for multisite metastasis, and 0% for bone metastasis. Overall, the five-year survival rate from the detection of distant metastasis was 52.9%, which is much higher than previously reported. This result might be consistent with those of Randle et al., who used data from the Surveillance, Epidemiology, and End Results registry to compare trends from 1983 to 2012 [32] and reported that disease-specific survival appeared to be improving in MTC patients with regional metastasis (from 82% to 91%) and distant metastasis (from 40% to 51%). In the present study, multiorgan metastasis and bone metastasis led to significantly poorer prognosis than lung metastasis (HR: 6.11 and 5.42, respectively).

Deaths resulting from DTC were rare, and many DTC patients passed from other causes [33]. Hence, the proportion of cancer-specific deaths in the MTC population was much higher than in the DTC population [8,28]. Although several studies have evaluated causes of death among DTC patients [34,35,36], few have evaluated those among MTC patients. In one study of 483 DTC patients who died between 1996 and 2018, Park et al. reported that only 16% of patients died of DTC, while the remaining 84% died from other causes [35]. In the present study, complications due to the progression of distant metastasis (36.8%) were the most common cause of death, followed by complications related to chemotherapy (26.3%) and airway obstruction due to tracheal invasion (21.1%). Complications due to the progression of distant metastasis included respiratory failure due to lung metastasis (15.8%), complications due to bone metastasis (15.8%), and mental changes due to brain metastasis (5.3%). All patients that died due to complications of conventional or targeted therapy died due to the progression of cachexia induced by chemotherapy. All patients complained of symptoms such as muscle wasting, anorexia, and reduced food intake during the end of the follow-up period, and their performance status was European Cooperative Oncology Group (ECOG) grade 4 during that period. The remaining 16% of patients died of MTC, but the direct cause of death could not be identified. Notably, two studies reported that respiratory failure and airway obstruction were the dominant causes of death in patients with papillary thyroid carcinoma, whereas complications due to immobilization arising from bone metastasis were the dominant cause of death in those with follicular thyroid carcinoma [35,36]. Among all causes of death, complications related to chemotherapy accounted for only 7.6% of deaths in the DTC population, whereas, in the present study, it accounted for 26.3%, demonstrating a significant difference [35].

The present study has several limitations. Primarily, it was conducted at a single tertiary referral hospital and designed retrospectively; thus, selection bias may be an issue. Second, we reviewed medical records and collected detailed information, but could not determine the direct cause of death in three deceased patients with MTC. Nevertheless, the present study included a large number of MTC patients who were followed for a long period of time. Additionally, long-term oncologic outcomes and detailed causes of death based on clinical data were identified in the present study.

## 5. Conclusions

In summary, distant metastasis was found in 19% of MTC patients during a mean follow-up period of 9.7 years, with one-third of cases confirmed at the initial diagnosis and two-thirds of cases confirmed during the follow-up period. There was no difference in the clinical characteristics observed according to the timing of distant metastasis. Overall CSS rates declined from 96% at one year to 63% at 10 years. Oncological outcomes differed by metastatic site; thus, patients with distant metastasis experienced a different clinical course. Patients with bone or multisite metastasis boasted significantly worse survival rates compared to those with lung metastasis. Complications due to the progression of distant metastasis were the most common cause of death, followed by complications related to chemotherapy and airway obstruction, respectively.

## Figures and Tables

**Figure 1 cancers-13-04670-f001:**
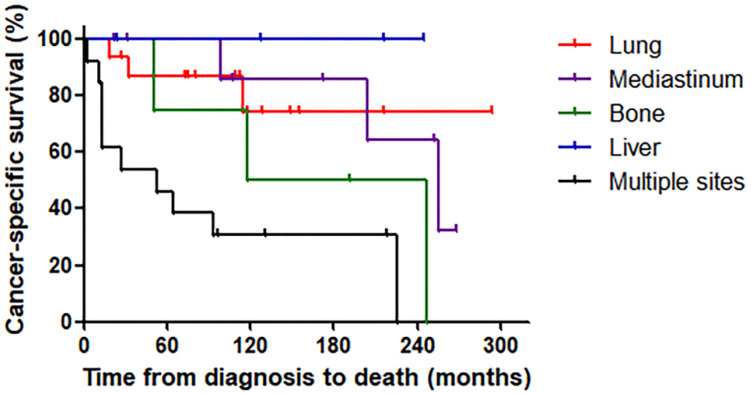
Cancer-specific survival according to initial distant metastasis site (log-rank *p* = 0.001).

**Figure 2 cancers-13-04670-f002:**
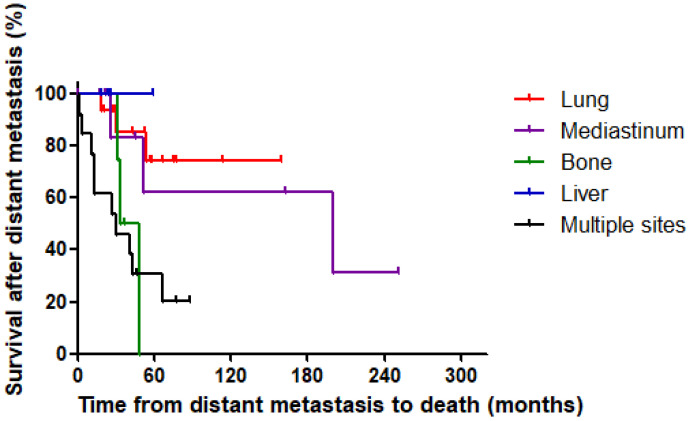
Survival after the detection of distant metastases according to the distant metastatic site (log-rank *p* = 0.007).

**Table 1 cancers-13-04670-t001:** Clinicopathological characteristics of 46 medullary thyroid carcinoma patients with distant metastasis.

Variables	Data
Age, years (mean ± SD)	45.9 (15.0)
Sex, women (*n*, %)	23 (50.0)
Tumor type (*n*, %)	
Sporadic	43 (93.5)
Hereditary (MEN 2A)	3 (6.5)
Primary tumor size, cm (median, IQR) †	3.0 (2.0–4.4)
Cervical lymph node metastasis † (*n*, %)	
N0/Nx	8 (17.3)
N1a	3 (6.5)
N1b	35 (76.1)
Extent of initial surgery (*n*, %)	
Total thyroidectomy	41 (89.1)
Lobectomy	1 (2.2)
Inoperable	4 (8.7)
Initial central lymph node dissection (*n*, %)	
Yes	41 (89.1)
No	1 (2.2)
Inoperable	4 (8.7)
Initial lateral lymph node dissection (*n*, %)	
Yes	34 (73.9)
No	8 (17.4)
Inoperable	4 (8.7)
8th AJCC/TNM stage † (*n*, %)	
Stage I	1 (2.2)
Stage II	3 (6.5)
Stage III	3 (6.5)
Stage IVA	20 (43.5)
Stage IVB	2 (4.3)
Stage IVC	15 (32.6)
No available data	2 (4.3)

AJCC, American Joint Committee on Cancer; IQR, interquartile range; SD, standard deviation; MEN, multiple endocrine neoplasia; SD, standard deviation; TNM, tumor–node–metastasis. † Primary tumor size and the presence of lymph node metastasis were assessed by image studies in inoperable patients, and no data were available on TNM stage in two patients.

**Table 2 cancers-13-04670-t002:** Comparison of clinical characteristics between SDM and MDM groups.

	SDM Group (*n* = 15)	MDM Group (*n* = 31)	*p*-Value
Age, years (mean ± SD)	47.3 (18.5)	45.3 (13.3)	0.675
Sex, women (*n*, %)	7 (46.7)	16 (51.6)	0.753
Tumor type (*n*, %)			
Sporadic	15 (100.0)	28 (90.3)	0.213
Hereditary (MEN 2A)	0 (0.0)	3 (9.7)	
Primary tumor size, cm (median, IQR) †	4.0 (2.6–5.0)	2.7 (1.8–4.0)	0.097
Cervical lymph node metastasis (*n*, %) †			
N0/Nx	2 (13.3)	6 (19.4)	0.635
N1a	0 (0.0)	3 (9.7)	
N1b	13 (86.7)	22 (71.0)	
Initial extent of surgery (*n*, %)			
Total thyroidectomy	11 (73.3)	30 (96.8)	0.008
Lobectomy	0 (0)	1 (3.2)	
Inoperable	4 (26.7)	0 (0.0)	
Initial central lymph node dissection (*n*, %)			
Yes	11 (73.3)	30 (96.8)	0.008
No	0 (0.0)	1 (3.2)	
Inoperable	4 (26.7)	0 (0.0)	
Initial lateral lymph node dissection (*n*, %)			
Yes	10 (66.7)	24 (77.4)	0.008
No	1 (6.7)	7 (22.6)	
Inoperable	4 (26.7)	0 (0.0)	
Number of dead patients	9 (60.0)	10 (32.3)	0.073

IQR, interquartile range; MDM, metachronous distant metastasis; MEN, multiple endocrine neoplasia; SD, standard deviation; SDM, synchronous distant metastasis. † Primary tumor size and the presence of lymph node metastasis were assessed by imaging studies in inoperable patients, and no data were available on TNM stage in two patients.

**Table 3 cancers-13-04670-t003:** Comparison of metastatic sites between synchronous distant metastasis and metachronous distant metastasis groups.

Metastatic Sites	Synchronous Distant Metastasis (*n*, %)	Metachronous Distant Metastasis (*n*, %)	Total (*n*, %)
Lung	6 (40.0)	10 (32.3)	16 (34.8)
Mediastinum	2 (13.3)	7 (22.6)	9 (19.6)
Bone	0 (0.0)	4 (12.9)	4 (8.7)
Liver	1 (6.7)	3 (9.7)	4 (8.7)
Multiple sites			
Lung + bone	2 (13.3)	2 (6.5)	4 (8.7)
Liver + bone	1 (6.7)	2 (6.5)	3 (6.5)
Lung + liver	1 (6.7)	0 (0.0)	1 (2.2)
Adrenal + lung	1 (6.7)	0 (0.0)	1 (2.2)
Adrenal + liver	0 (0.0)	1 (3.2)	1 (2.2)
Brain + lung	0 (0.0)	1 (3.2)	1 (2.2)
Brain + pancreas + lung + liver + bone	1 (6.7)	0 (0.0)	1 (2.2)
Kidney + liver + bone	0 (0.0)	1 (3.2)	1 (2.2)

**Table 4 cancers-13-04670-t004:** Survival rates from the detection of distant metastasis and HRs according to metastatic sites.

Site of Metastasis	Number of Patients	Number of Cancer-Specific Deaths	5-Year Survival from the Detection of Distant Metastasis (%)	HR (95% CI)	*p*-Value
Overall	46	19	52.9		
Lung	16	3	74.6	Reference	
Mediastinum	9	3	62.5	1.43 (0.24–8.54)	0.679
Bone	4	3	0.0	5.42 (1.05–27.99)	0.044
Liver	4	0	100.0	-	-
Multiple sites	13	10	30.8	6.11 (1.67–22.34)	0.006

HR, hazard ratio; CI, confidential interval.

**Table 5 cancers-13-04670-t005:** Detailed information of 19 deceased patients.

No.	Age	Sex	RET Mutation	8th TNM Stage	Site of Distant Metastasis	Cause of Death	Duration (Months)
1	56	M	NA	Not available	**Lung, bone**	Airway obstruction	13
2	60	F	NA	IVC	**Lung, adrenal**	Airway obstruction	3
3	25	F	NA	IVC	**Lung**	Airway obstruction	19
4	48	M	Wild-type	III	**Mediastinum**	Airway obstruction	99
5	35	M	Wild-type	IVA	**Lung**, bone, liver	Respiratory failure	115
6	60	M	NA	IVC	**Lung**, bone	Respiratory failure	33
7	41	M	NA	IVA	**Mediastinal soft tissue**, lung, pleural	Respiratory failure	255
8	34	M	Wild-type	IVC	**Lung, bone**	Complications due to bone metastasis	247
9	46	F	Wild-type	III	**Bone**	Complications due to bone metastasis	11
10	57	M	Wild-type	II	**Bone, liver**	Complications due to bone metastasis	93
11	65	F	Wild-type	IVC	**Lung, bone, liver, pancreas**, brain	Complications due to brain metastasis	65
12	77	F	NA	IVC	**Bone, liver**	Complications related to conventional chemotherapy (capecitabine)	13
13	63	M	NA	IVC	**Mediastinal soft tissue**, bone	Complications related to targeted therapy (vandetanib)	204
14	64	M	NA	IVA	**Lung, bone**, liver	Complications related to targeted therapy (vandetanib)	13
15	55	M	Wild-type	IVC	**Lung, liver**, bone, pancreas, brain	Complications related to targeted therapy (lenvatinib)	118
16	46	M	Wild-type	Not available	**Bone**, lung, pleural, pancreas	Complications related to targeted therapy (vandetanib)	226
17	54	M	Wild-type	IVA	**Bone**	Unspecified	51
18	24	F	Wild-type	IVC	**Bone, lung**	Unspecified	27
19	71	M	Wild-type	IVA	**Liver, kidney**, bone	Unspecified	53

Duration, period from initial diagnosis to death (months); NA, not assessed; TNM, tumor–node–metastasis. Bold text represents initial metastasis site.

## Data Availability

The data presented in this study are available on reasonable request from the corresponding author.

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
