# Peer review of "Long-Term Outcomes and Causes of Death among Medullary Thyroid Carcinoma Patients with Distant Metastases"

_cancers, 2021, doi:10.3390/cancers13184670_

Round 1
Reviewer 1 Report
In table 1, it is reported that 42 patients underwent surgery. Of these, 35 were N1b; for these patients, the surgery includes lateral neck dissection; this data should be reported.
Was the RET gene mutation investigated even in sporadic cases? Was there a correlation between, if any, mutation and tumour aggressiveness?
Both Ct-DT and CEA-DT were demonstrated to be significant predictors of prognosis in MTC. Was the Calcitonin and CEA doubling time evaluated? Was this data considered regarding the overall survival?
Also, the tumour volume doubling time has been reported as a strong predictor of OS in patients with recurrent or metastatic MTC. Was a correlation made between the structural doubling time of the tumour and OS in this series?
Reviewer 2 Report
The authors evaluated prognosis in patients with metastatic medullary thyroid cancer (MTC). They found survival outcomes differed depending on metastatic sites. MTC patients with bone metastasis and multisite metastasis had poorer survival than those who had lung metastasis. Among patients died of MTC, complications associated with distant metastasis were the most common cause of death, followed by complications related to chemotherapy and airway obstruction. There are some comments on this draft.
- In Table 2, SDM group, many of these patients (73.3%) received total thyroidectomy. The authors may clarify the reasons why total thyroidectomy was performed in patients who had metastatic MTC already.
- In Table 3, the percentages were not given in many groups, including bone + lung, liver + bone etc., which may be in included in this Table.
- A new figure of cancer-specific survival curve according to distant metastasis sites from the detection of distant metastasis may be included in this draft. This figure will help to assess the effects of various metastatic sites on survival since metastasis appeared.
- In Table 5, the genetic analysis may specify the genetic alterations, such as RET, HRAS, and KRAS, rather than sporadic.
- There were 19 out of 46 patients (41.3%) died of MTC in this study. It wound be important to known the risk factors associated with mortality. The data would help clinician identify MTC patients who are particularly at risk of death from this disease.
- In Table 5, patient #12 died of complication associated with chemotherapy. The regimen of chemotherapy may be included in this Figure.
- In Table 5, vandetanib and lenvatinib may be more appropriate to be termed as targeted therapy, rather than chemotherapy.
- In Table 5, five patients died due to complications of treatment. The authors may address these complications. These data might give some hints to improve the management for MTC patients in the future
- In Table 4, fiver year survival was 0% in patients with bone metastasis. The authors may clarify if these patients received treatment for bone metastasis, such as radiotherapy or antiresorptive therapy.
- In Table 5, the unit of duration should be included.
Round 2
Reviewer 1 Report
The Authors edited the manuscript with respect of the reviewer's suggestion. The manuscript was improved.
Reviewer 2 Report
The quality of this draft has been substantially improved after revision. The authors have largely adequately addressed my questions. However, there are still some minor comments on this draft.
- In line 13, “metastasis to death, and we found long-term oncologic outcomes were differ from initial metastatic” may be “metastasis to death, and we found long-term oncologic outcomes were different from initial metastatic”.
- In line 27, “two groups except regarding initial surgical extent” may be “two groups except initial surgical extent”.
- In line 111, “their first operation” may be “the first operation”.
- In line 158, “analyze the factors affecting” may be “analyzing the factors affecting”.
- In lines 298-300, this sentence needs to be revised to be readable.